# Effect of Rice Processing towards Lower Rapidly Available Glucose (RAG) Favors *Idli*, a South Indian Fermented Food Suitable for Diabetic Patients

**DOI:** 10.3390/nu11071497

**Published:** 2019-06-30

**Authors:** Ramachandran Chelliah, Sangeeta Chandrashekar, Kandasamy Saravanakumar, Sudha Rani Ramakrishnan, Momna Rubab, Eric Banan-Mwine Daliri, Kaliyan Barathikannan, Akanksha Tyagi, Fred Kwame Ofosu, Xiuqin Chen, Se-Hun Kim, Fazle Elahi, Han NaKyeong, Myeong-Hyeon Wang, Vijaykumar Raman, Usha Antony, Deog-Hwan Oh

**Affiliations:** 1Department of Food Science and Biotechnology, College of Agriculture and Life Science, Kangwon National University, Chuncheon 24341, Korea; 2Department of Physiology, Bharath Institute of Higher Education and Research, Chennai 600 073, India; 3Department of Medical Biotechnology, College of Biomedical Sciences, Kangwon National University, Chuncheon 24341, Korea; 4School of Food Science and Biotechnology, Kyungpook National University, Daegu 41566, Korea; 5Department of Biotechnology and food technology, Anna University, Chennai 600 025, India

**Keywords:** rice, *Idli/*Dokala—Indian food product, glycemic index, heat treatment, RAG, SAG, glucose level

## Abstract

The Asian food pattern primarily embraces rice and rice-based products, which mainly comprise 90% starch. Among these various food products, *Idli* is mostly prepared through fermentation. It has high palatability, and the rapid and highly digestible nature of the food product results in a higher post-glucose level in diabetic patients. A heat-treated *Idli* rice sample was prepared by roasting parboiled rice at the temperature range of 155 to 165 °C for 65 to 75 s. *Idli*/rice-based Dokala made from heat-treated rice is better when compared to untreated rice in terms of its microbiological profile and physiochemical properties. The proximate composition of heat-treated parboiled rice *Idli*/Rice Dokala showed slightly higher values than the untreated parboiled rice *Idli*/Rice Dokala, which reveals that it has marginally higher nutritive value. Determination of the Rapidly Available Glucose (RAG) and Slowly Available Glucose (SAG) values, SEM analysis, resistant starch analysis, microbial assay, and in vivo studies were performed to determine the glycemic index (GI) and glycemic load in normal and diabetic persons. Sensory analysis also proved that heat-treated *Idli*/Rice Dokala is superior to untreated based on the color, flavor, appearance, taste, and texture.

## 1. Introduction

Diabetes has developed into a chronic disease in humans, affecting approximately 200 million adults globally and almost 15 million Americans in 2000. In 2002, the rate of diabetes increased drastically; also, the treatment and medicine productivity costs were raised. Likewise, the incidence of diabetes among U.S. adults has grown to 61%; previously, it was 5% in 1990, and it continuously increased to 8.1% in 2000 [1]. It has been predicted that in 2075, diabetes will affect almost 29 million adults in the United States. The lifetime risk of diabetes based on gender was 32.8% in males and 39.2% in the female population of the United States in 2000 [2]. Moreover, the extreme level of diabetes-affected persons led to an increased death rate, accounting for 71,372 deaths in 2000 [3]. It has been anticipated that an examination for diabetes at 50 years of age will result in increased life expectancy for about 12.5% of males and 15.1% of females [2] (Figure 1). 

Famous traditional rice-based Indian foods such as *Idli* and Dokala are mainly prepared with parboiled rice, which is rich in amylose content, leading to soft and spongy textural characteristics [4]. Fermentation and cooking cause a reduction in oligosaccharides of 40% and result in a reduction in levelness of the product based on the substitution of other grains such as pulses (kidney beans, black gram, broad bean, chickpea, soybean, and lentil); hence, depending on the composition, the starch digestibility—indirectly responsible for our understanding of the context of type 2 diabetes—differs [5]. Several studies have been conducted to understand the blood glucose response based on the carbohydrate content of food products. Usually, blood glucose is quantified based on the glycemic index (GI). Apart from the glycemic index, the resistant starch (RS) content is measured based on starch digestibility. The fabrication of gradually digestible starch (GDS) and RS has been established to alleviate the quantity of blood sugar and to produce aids to human health [6]. The RS level of rice is increased by heat treatment methods, and it has been reported that high-pressure heating (116 °C, 10 min) increased the RS level in raw rice from 6.29% to 14%. Nevertheless, parboiling reduced the RS by 50% in long-grain rice. It appears that heating at higher than 100 °C possibly enhances the RS content. Moist heat treatment (MHT) is a hydrothermal method which changes the physiochemical properties of starch and is mostly performed at a controlled humidity level (lower than 35%) and by heating at 84–140 °C for 15 min to 16 h [7]. It leads to gelatinization of the rice product due to a lower moisture content. Also, changes such as the development of complexes of amylose–lipid (ALC) and lowered enzymatic vulnerability occur during MHT [8]. The quantity of RS in MHT-treated starch leads to breakdown of the granules’ minor structure. The assembly of gradually digestible starch (GDS) and RS has been initiated to regulate the quantity of blood sugar towards better health and welfare [6].

## 2. Dietary Factors and Type 2 Diabetes

The complex nature of inter-connections between nutrients and nourishment in the diet and the hazard of diabetes and high glucose levels in the blood has been inspected by numerous researchers. The most prominent research indicates that a lifestyle of consuming a high-fat and high-calorie diet leads to type 2 diabetes, but epidemiologic research information has shown that this is relatively unpredictable [9]. Recent studies have revealed that specific food intake based on high glucose content and carbohydrates and an irregular dietary profile [10] leads a person to be more prone to diabetes.

## 3. Contemporary Perceptions of the Deterrent Levels of Starch/Carbohydrate/Fiber/Fat in Diet

Currently, the emphasis being placed on the glycemic reaction to various carbohydrate-enhanced foods varies from the conventional conviction about basic versus complex carbohydrates in the management of diabetes mellitus [11]. Recently, significant evidence has shown that different carbohydrate-containing foods reveal the same quantity of sugars in chemical analysis but cause wide variation in blood glucose tolerance or response after the ingestion of food.

## 4. Rice and the Glycemic Index Mechanism

Rice, a staple food in India, is a prime source of carbohydrate, and different varieties are applied as a primary food. However, the GI of rice is higher than that of any other starch-containing food [11]. Rice can be biochemically translated as starch (60%–80%). The digestion of the starch is carried out by the enzymes amylase and amyloglucosidase. Alpha-amylase (1, 4 α D-Glucan-glucanohydrolase) targets outsized polymers of starch at intimate bonds and splits them into short glucose polymers [12]. Amyloglucosidase catalyzes the hydrolysis of the alpha-1, 4 and alpha-1, 6 linkages of starch, producing glucose. During hydrolysis, glucose is removed from the non-reducing end of amylose or amylopectin chains [13,14,15]. The basic hydrolysis rate depends on the type of linkage and chain length: for example, alpha-1, 4 linkages are more readily hydrolyzed than alpha-1, 6 linkages [16]. The amyloglucosidase mechanism focuses on the shorter polymers and splices them off into single glucose units (Figure 2). 

## 5. Glycemic Index and Glycemic Load

Relatively less epidemiologic research-based evidence has shown the correlation of carbohydrates with diabetes to be indirectly proportional to high glucose in the blood [17,18,19,20]. However, groups of people consuming low levels of carbohydrate or simple sugars such as monosaccharide or disaccharide show lower incidence of diabetes [20,21,22,23]. An exception, the Iowa Women’s Health Study, indicated that the uptake of fructose or glucose leads to the risk of stimulating type 2 diabetes mellitus [24,25,26]. With the development of the GI in 1980, it was concluded that the relation of carbohydrates with type 2 diabetes depends on the structure [27,28,29]. The GI matches the glucose uptake, which is proportionally equal to the absorption of sugar in one’s daily diet [27]. In general, high glucose and high GI uptake lead to type 2 diabetes [29]. 

The American Diabetes Association (ADA) endorses the consumption of nutritious food such as fruits, vegetable, cereals, pulses, and dairy products as a healthy food diet for diabetes patients, because lower amounts of carbohydrate lead to lower incidence of diabetes [30]. The ADA specifically shows instructions for estimating the uptake of glucose in blood from the specific level of uptake of carbohydrate in a meal or per day [31]. Numerous factors such as the nature of the sugar; the quantity of fiber, lipid, and protein; the food form; the various methods of preparation; and the significant role of enzyme inhibitors influence the GI of a specific food.

The current research shows that higher-carbohydrate/fiber diets result in a reduction in blood glucose and other beneficial factors towards controlling diabetes, specifically in those with type 2 diabetes mellitus [31,32,33]. Similar foods that possess lower GI correspondingly assist in reducing blood cholesterol and triglyceride points. Such foods constitute the main quota of mono, di, and polysaccharides responsible for diabetes. An extended period of research has indicated that the GI is a valuable concept, and the application of sucrose in a reasonable quantity does not affect glucose control [33]. This is a step towards understanding the difference between starch-enriched diets and carbohydrate-enriched nourishment or monosaccharide versus polysaccharide [32] (Figure 3).

## 6. The Roles of Rapidly Available Glucose (RAG) and Slowly Available Glucose (SAG) in Diabetic Patients

The final product of the digestion of starch is glucose, which is classified into Rapidly Available Glucose (RAG), Slowly Available Glucose (SAG), Free Glucose (FG), and Total Glucose (TG) [34]. RAG and SAG reflect the rates at which sugars (glucose and starch, including maltodextrins) become available for the absorption in the human gut [29]. RAG is rapidly unrestricted and absorbed (within 30 minutes) and is a key factor of glycemic response, while SAG is unconstrained and absorbed slowly and is not expected to affect the glycemic response. Reduced-GI foods yield a gradual increase in blood sugar (RAG) and are associated with reduced risk of diabetes, cardiovascular disease, and cancer [35]. Foods that are more slowly digested may have metabolic benefits in diabetes because the RAG values for such foods would be low and they would not result in a rapid postprandial increase in glucose level. This is beneficial in the dietary management of diabetic patients (Figure 3). 

## 7. Resistant Starch and Gut Microbiota Modulation

Resistant starch (RS) is well defined as the starch and products of starch degradation that are not absorbed in the small intestine of healthy individuals [38]. RS resists digestion by enzymes in the digestive tract but can be partially broken down in the large intestine by gut bacteria. Carbohydrates are made up of sugars, starches, and fibers (Table 1) [31]. In the colon, they are moderately fermented by gut bacteria in a way similar to digestible fibers. The structure of the human microbiome, altered by comparatively few dietary constituents, has been inspected for various functional potentials of the microbiome, such as an increase in Firmicutes to Bacteroidetes leading to an increase in enzymatic pathways involved in phospholipid metabolism in the gut (Figure 4).

The proportion of amylose and amylopectin, physiochemical form, degree of gelatinization, heat treatments, refrigeration, and storage affect the RS quantity in foods [40,41,42]. RS in nutraceutical-based food processing and manufacturing has drawn attention, leading to wide-ranging inquiry into the influence of RS on the non-digestible sugar component of the diet and its physiological implications. RS may be made up of retrograded starch, physically inaccessible starch, starch–nutrient complexes, chemically modified starch, and starches that are complex due to enzymatic inhibition [43,44]. The RAG and SAG values could be altered by RS formation (Figure 5).

As recorded, there are four types of RS: RS1 is starch which is physically difficult to reach due to the manifestation of rigid cell walls in grains, kernels, or rhizomes. RS2 is native crystalline starch granules that are resistant to enzymatic hydrolysis; for example, the starch in uncooked potatoes, the green variety of banana, and elevated-amylose corn. RS3 is retrograded starch which is cooked followed by cooling for a few hours or longer. RS4 is the chemically changed starches; these are due to cross-linking with substance reagents, ethers, esters, and so forth [45].

## 8. Resistant Starch (RS), Insulin, Glucose Metabolism, Glycemic Load (GL), and Glycemic Index (GI)

Insulin is considered to be the hormone accountable for the transportation of blood-based glucose from the circulatory system to the muscle and (fat) cells [46]. Avoiding insulin hikes has been observed as the primary feature in managing body weight as this hormone not only stores blood-based glucose in lipid-based cells but additionally blocks the arrival of lipids and their subsequent use for energy. Foods containing RS cause continuous insulin discharge; this causes the eradication of lipids to be elevated in the liver and is ultimately utilized for energy (Figure 6). Accordingly, RS assumes a critical role in the insulin reaction balance and in managing glucose digestion [32,47]. As an outcome, the use of RS-rich nourishment is imperative in the management of body weight and obesity and in diminishing the danger of developing diabetes and other significant illnesses. 

## 9. Fermented Foods—*Idli*

Fermented foods are often more nutritious than their unfermented counterparts, and they are easily digestible. This is true of naturally fermented legume products; hence, such foods are still produced in India by simple traditional methods [49]. *Idli* is the most common fermented legume and rice product in South India. *Idli* is highly digestible and is thus recommended as a food for patients, infants, and older people with impaired digestion [49]. However, diabetic patients are advised to avoid *Idli* because of its rapid digestibility, which in turn results in a rapid rise in their postprandial blood glucose level. This is detrimental to diabetic patients [50]. The GI of *Idli* made in the traditional method is as follows, making *Idli* not suitable for the diabetic diet. 

## 10. Limitations of the Glycemic Index (over 77 ± 2, Considered High) and the Glycemic Load (over 40.04, Considered Hiked) for a Serving Size of 250 g

In order to make *Idli* suitable for diabetic patients, the digestibility must be reduced to give it lower RAG values [51]. This will result in a lower GI. So far, little work has been published on the development of modified starch products. Therefore, the objective of the present work follows Phase I, which involves the optimization of the heat treatment of rice. 

## 11. Processing Method

In food processing, physio-chemical treatments may meaningfully affect starch ingestion. For grain-based foods, modification based on dampness, condition, and so forth is vital [52]. Starch is a combination of binary forms such as amylose and amylopectin, both of which are polymers of glucose. In terms of human nutrition, starch is by far the most important of the polysaccharides. The range of change is typically estimated from dextrose correspondents, which are crudely the segments of the glycoside bonds in the starch which have been fragmented [52,53]. 

## 12. Physical Treatment

Moisture–heat action has been generally considered a main strategy for altering starch yet heat treatment of starch in a dehydrated environment has been rarely examined. One exemption is pyrodextrin creation by the treatment of starch at high temperatures [54]. Dextrins can be fashioned from starch based on enzymes like amylases by ingestion inside the human body and during malting, squashing, or by applying a high temperature under acidic conditions (pyrolysis or roasting) [55]. It has been claimed that heating starch or flour at >100 °C, preferably for numerous hours, produces functionality equal to that obtained by chemical cross-linking [56]. The rice assortment-based heat discharge process is a novel system for gelatinizing crude starch (Figure 7). It exploits warmed air to substitute for steam when preparing rice under high-temperature fluidization over a brief period. The dextrin shape is difficult to denaturalize and simple to store in contrast with that created by conventional aquatic-based steaming [57]. 

## 13. Microstructure of Rice Grains

Eliasson et al. [57] reported that dehydrated warming could be used as an alternative procedure to the biochemical alteration commercially applied to starches of different origins. Starch under thermo-treatment at 100 °C generated a well-designed equivalent to that obtained by chemical cross-networking by increasing the resistance to thickness (viscosity) breakdown and by shortening the glued structure [59]. Mild heat can be used to change the physical abilities of starch [54,55,56,57]. Moist heat treatment (MHT), which involves heating at a controlled humidity level, is a mono process applied to substantially alter starches and rice flours [57]. Eliasson et al. [57] observed that the physical and chemical properties of starches were radically altered after MHT [52,59]. Enhanced RS contents and reduced enzyme vulnerability of starches may be accomplished by different time/temperature treatments; this has been shown previously for starches of different botanical origin and with varying amylose/amylopectin ratios [48,60,61,62,63,64,65,66,67,68]. Dehydrated heating (130 °C) of waxy corn starch expanded the GDS content. Among the temperatures utilized, 130 °C was found to increase the SDS content [69] (Table 2). 

## 14. Glycemic Index

The GI concept was initially developed to categorize diverse sources of sugars and starch-rich foods in the diet based on their influence on postprandial glycemia [70,71,72,73]. Reduced-GI foods are known to produce less postprandial hyperglycemia and hyperinsulinemia than high-GI foods [67,73]. The GL helps quantify the distinct effect of a food product with readily obtainable carbohydrate. In other words, the GL is more practical than the GI of food items in terms of measuring glucose in the circulation [GL = (GI × dietary carbohydrate of serving)/100] (Table 3).

An ADA spot report on nourishment concluded that the application of GI and GL would possibly give greater advantage in regulating diabetes based on total carbohydrate [30]. Additionally, the ADA stated that the data are not adequate to permit a decision on whether low-GL foods reduce the risk of diabetes; however, the ingestion of reduced-GI foods that are high in iron and fiber fortified is recommended [30]. An elevated association exists between the degree of in vitro glucose increase from starch-rich nourishments, centered on pancreatic and digestive enzymes, and the glycemic response in vivo [74]. Numerous dietary factors are directly proportional to glycemic and insulin responses, the carbohydrate (monosaccharide, disaccharide, and polysaccharide) uptake and ingestion rate, and the rate of secretion of digestive enzymes [48,66,67,68]. The rate of breakdown of nutrients differs relative to the physiochemical properties of the food [72,73,74].

Table 3 indicates the hydrolysis index (HI) and assessed GI values of the processed rice samples. Overall, the HI was enhanced with an increase in the degree of cooking of the rice (R < MH < CC), while the enhancement of MH and CC for non-homogenized samples was not statistically significant. Chung, Lim, and Lim [8] reported an absence of variation in starch hydrolysis of sticky rice for samples of diverse gelatinization grades. 

## 15. Rapidly Available Glucose (RAG) and Slowly Available Glucose (SAG)

The glycemic sugar fraction, which is captured in the small intestine, is defined as the quantity of carbohydrate and starch excluding Resistant Starch (RS). The sugar fraction can be subdivided into Readily Available Glucose (RAG) and Slowly Available Glucose (SAG), and in vitro analysis reflects the likely degree of absorption of glucose in the small intestine. Analytical procedures have been developed to characterize foods and reflect their expected biological uptake. The values of RAG and SAG were determined in hydrolyzed foods by enzymes amyloglucosidase and amylase at 20 minutes and 120 minutes of an in vitro carbohydrate digestion method, respectively, by modifying the method by Eliasson [57]. The RAG index was confirmed as a forecaster of the potential glycemic response obtained from the digestion of these food items. The hypothesis is that the RAG fraction is rapidly (within 20 minutes) released and absorbed and is the major concern of the glycemic reaction. The SAG fraction is unconstrained and absorbed slowly and is not anticipated to affect the glycemic response. The in vitro RAG of foods could imitate the glycemic reaction [59]. Fredriksson indicated that in vitro estimation of the carbohydrate ingestion rate is correlated with the glycemic response in human study; this demonstrates that in vitro carbohydrate ingestion can be a valuable model to estimate the biotic response of a carbohydrate-based meal while also being a simple and inexpensive method [69]. 

## 16. Processing Method and Resistant Starch

In food processing, physio-chemical treatments may profoundly affect starch indigestibility. For cereal foods, modifications in humidity, thermal conditions, etc., have been verified [52]. Starch is a combination of binary factors such as amylose and amylopectin, both of which are polymers of glucose. In terms of human nourishment, starch is by far the most significant of the polysaccharides. The level of adaptation is typically estimated from dextrose correspondents, which are coarsely the segments of the glycoside bonds in the starch which have been broken [56]. Resistant starch (RS) is considered one of the three types of dietary fiber, along with the carbohydrate that is digested in the small intestine, and that which reaches the large intestine undigested [69].

Further bacterial fermentation leads to the production of volatile fatty acids. This has been observed in the majority of carbohydrate-enriched foods. Consumption of natural RS by humans has been indicated to result in a reduced glycemic response in healthy individuals [35], decreased glycemic response in diabetes, and increased insulin sensitivity in healthy individuals [35,38,75] (Table 4). 

An important association was detected when the percentage of rapid carbohydrate ingestion and the Lente carbohydrate assimilation percentage were interrelated with the GI of the food consumed, but only when a single rapid carbohydrate absorption rate was considered. These outcomes validate this as a beneficial, simple, and low-cost method to evaluate the biotic response of high-carbohydrate foods [68,69]. 

## 17. Conclusions

The review showed that a slight amendment to rice treatment can modify the physiochemical properties of rice. This rice provides bulk, that is, to satiety, but most is not absorbed and it does not lead to an increase in blood glucose level. By the additional effect of fiber and further effects as discussed, overall and Low-density lipoprotein (LDL) dietary fat was reduced. The process is economically affordable and not expensive. The only disadvantage is minor differences in taste and the raised fuel rate required for heating the rice. Based on this review, it was concluded that the thermal treatment of rice is a better method for increasing the resistant starch content, which, in turn, may be useful for diabetes patients. Hence, heat treatment of rice is preferred for the preparation of low-GI *Idli* and other rice-based food products. 

## Figures and Tables

**Figure 1 nutrients-11-01497-f001:**
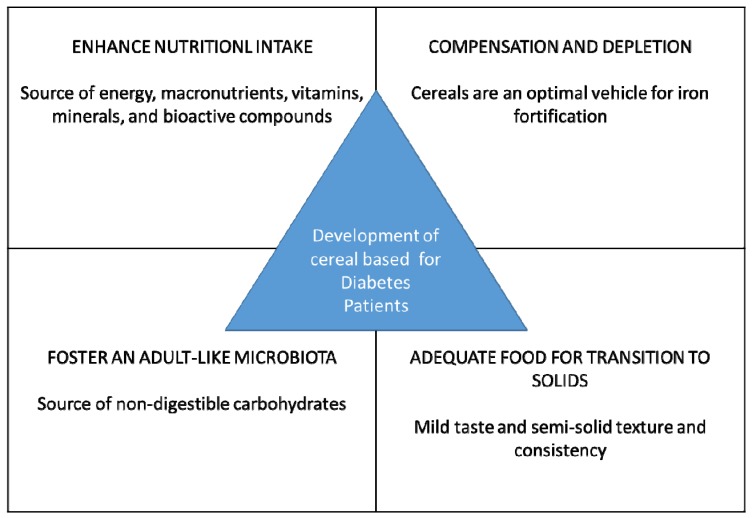
Development of cereal-based products for diabetes patients.

**Figure 2 nutrients-11-01497-f002:**
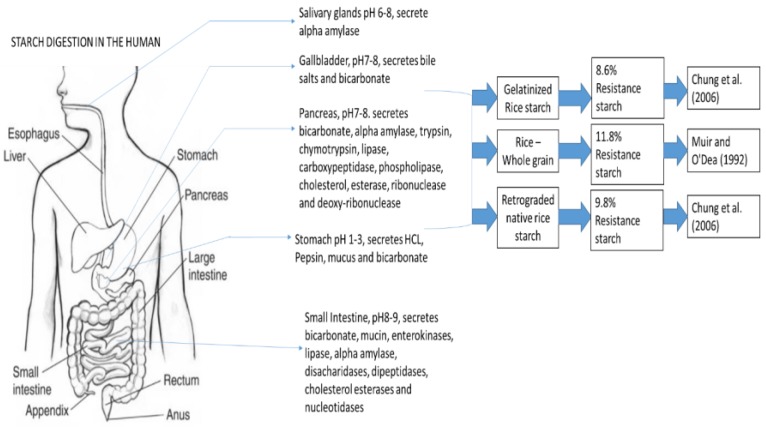
Digestive enzymes secreted by different organs contribute to the digestion of foods in humans. The digestive system, an extension of the environment into the body, also stores mucus, acid, bicarbonate, and salts to enable hydrolysis of foods and absorption of nutrients essential for life.

**Figure 3 nutrients-11-01497-f003:**
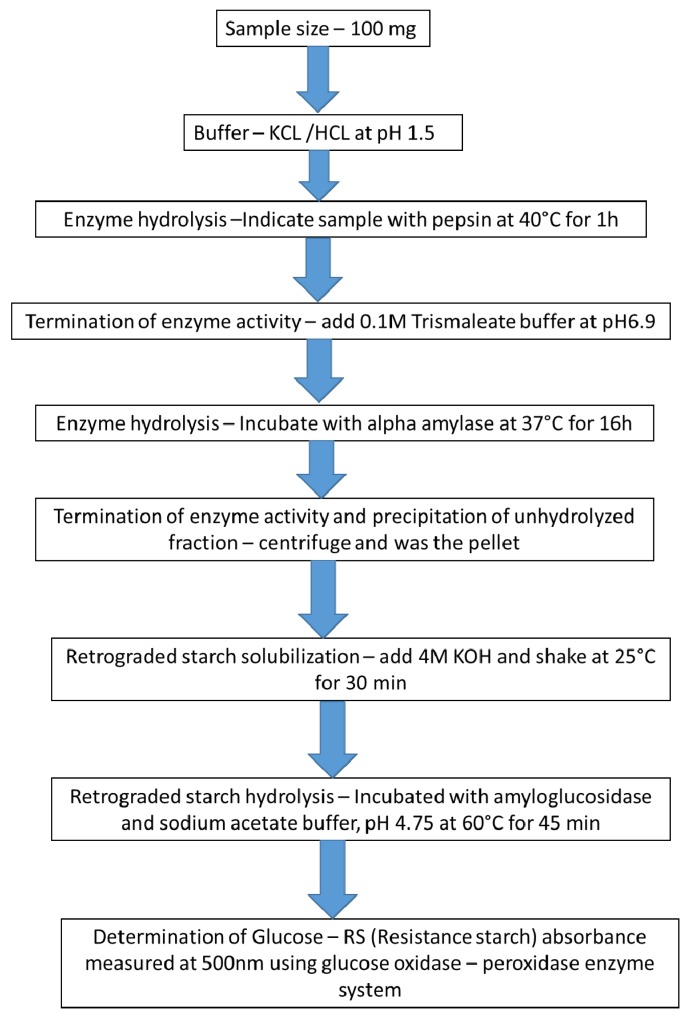
The procedure used by Goñi et al. [36] and Perera [37] for the determination of resistant starch (RS) in food. Copyright obtained from Perera et al. [37].

**Figure 4 nutrients-11-01497-f004:**
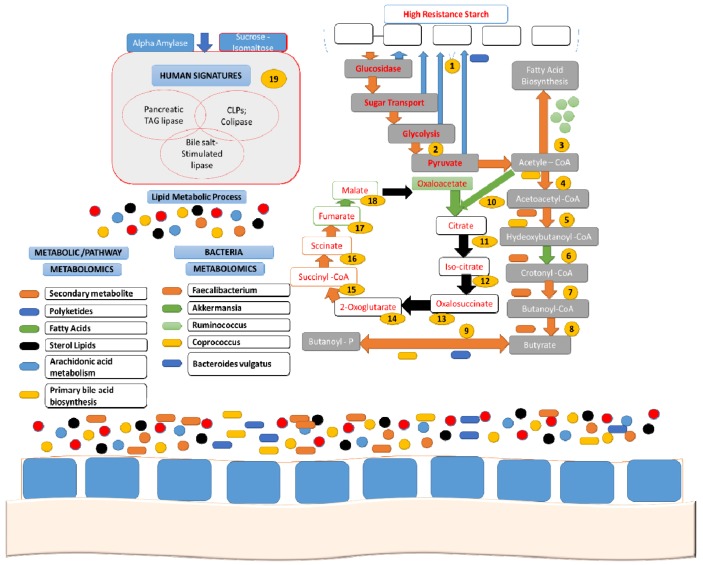
Summary of distinguished enzymes, subsequent pathways, particular species, and specific metabolites that are expressively obstructed through a high resistant starch (HRS) diet. Blue arrows/frames, increased with HRS; orange frames, decreased with HRS; black arrows, not detected or not increased with HRS over baseline; green arrows/frames, increased with low resistant starch (LRS). 1, starch and metabolism of sucrose; 2, glucose to pyruvate leads to glycolysis pathway; 3, 3-oxoacyl-(acyl carrier protein) synthase; 4, acetyltransferase-based acetyl-CoA; 5, 3-hydroxyacyl-CoA dehydrogenase; 6, enoyl-CoA hydratase; 7, carrier protein enoyl-(acyl) reductase (NADH); 8, acetate transferase CoA; 9, butyrate kinase; 10, synthase citrate; 11, hydrataseaconitate; 12 and 13, dehydrogenase isocitrate; 14, Ferredoxin oxidoreductase-2-ketoglutarate; 15, CoA-synthetase-succinyl; 16, fumaratereductase/succinate dehydrogenase; 17, fumaratehydratase; 18, dehydrogenase malate; 19, human enzymes.

**Figure 5 nutrients-11-01497-f005:**
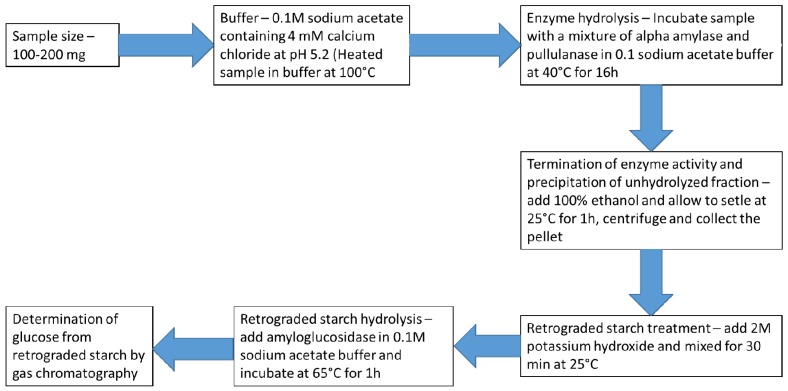
Important steps according to the protocol by Englyst et al. [43] towards determining enzyme-resistant starch (RS3). The manifestation of an immune enzyme fraction of starch in foods during the measurement of non-starch polysaccharides in foods. Copyright obtained from Perera et al. [37].

**Figure 6 nutrients-11-01497-f006:**
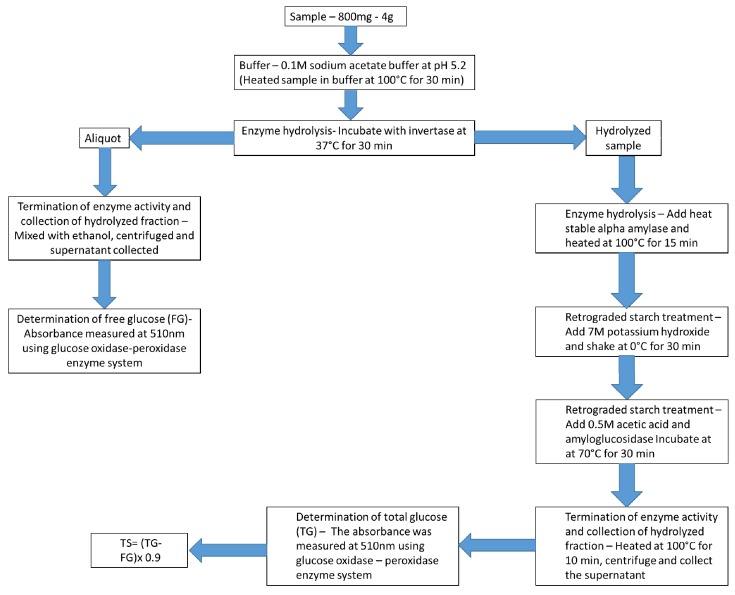
Instantaneous procedure for the determination of total starch (TS) as per Englyst et al. [48]. The freely available glucose (FG) and total glucose (TG) matters were defined by quantifying the total starch content of the sample. Copyright was obtained from Perera et al. [37].

**Figure 7 nutrients-11-01497-f007:**
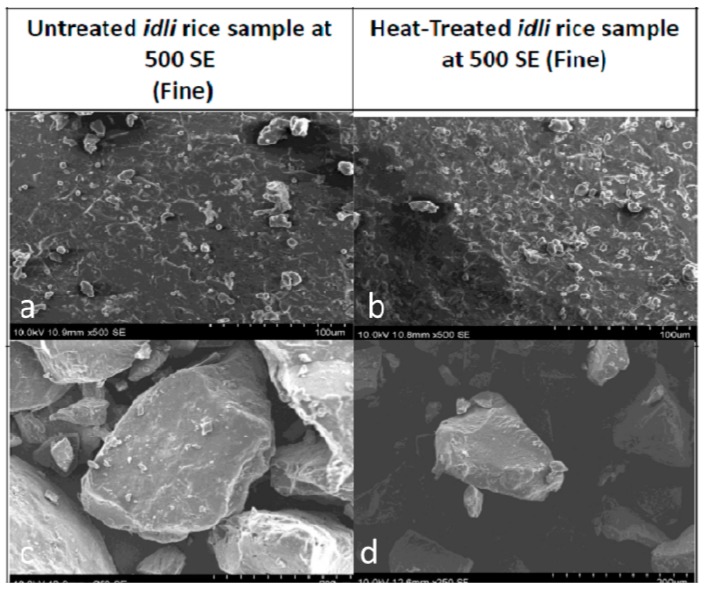
Transmission electron microscopic images indicate the effect of heat treatment on the prepared *Idli* rice product compared with the control *Idli* (No Heat Treatment) [58]. (**a**) The fine particle size of raw rice (Magnification at 100 μm); (**b**) heat-treated rice particles with increased starch content (Magnification at 100 μm); (**c**) the rough surfaces of the raw rice particles (Magnification at 200 μm); (**d**) heat-treated rice particles with increased starch show smooth surfaces (Magnification at 200 μm).

**Table 1 nutrients-11-01497-t001:** Estimated total dietary fiber, total starch, and resistant starch of some nourishment sources (g/100 g as eaten).

Source	Total Starch	Total Dietary Fiber	Resistant Starch
**Legumes**			
Red beans	42.4	36.2	24.4
Pulses	53.2	33.1	25.3
*Vigna unguiculata* (Peas)	53.6	32.6	17.5
**Cereal grains**			
Barley (*Hordeum vulgare*)	55.2	17	18.3
Corn Starch	77.5	19.3	25.7
Arborio rice	95.2	1.2	14.2
Wheat	50.4	17.2	13.2
Oats	43.2	37.2	7.2
**Flours**			
Corn	84.2	2.3	11.3
Wheat	68.5	12.6	1.4
Rice	86.3	5.4	1.5
Potato	81	2.7	1.6
**Grain-based food products**			
Spaghetti	73.9	5.7	3.1
Rolled oats	56.4	10.3	8.2
**Cereal products**			
Crisp bread	67.2	N/A	1.4
White bread	46.3	N/A	1.6
Granary bread	44.6	N/A	6.2
Extruded oat cereal	57.1	N/A	0.5
Puffed wheat cereal	67.7	N/A	1.5
Oat porridge	9.2	N/A	0.4
Cooked spaghetti	N/A	N/A	2.7
Cooked rice	N/A	N/A	3.8
**Potato products**			
Boiled potatoes	N/A	N/A	2.1
Chips	29.1	N/A	4.5
Mashed potatoes	N/A	N/A	2.1

Quantification of total starch, total dietary fiber, and resistant starch levels in diverse raw food material and expressed in terms of percentage. Copyright obtained from Tamura et al. [39].

**Table 2 nutrients-11-01497-t002:** Preparation of resistant starch in starch-rich products by different treatments.

Starch Source	Treatment	Resistant Starch (%)	Reference
Rice starches	Native	6.1–10.4	
	Heat–moisture treatment (HMT)	18.3–23.1	
	Acid and HMT	30.1–38	
Corn	Native	4.3	[48]
	Autoclaving	24–31	
	a-amylase and pullulanase	58.83	[8]
	Partial acid hydrolysis followed by HMT	63.2	[8]
Potato starch	Raw	75	
Wheat starch	Cross-linking with sodium trimetaphosphate sodium tripolyphosphate, epichlorohydrinandphosphoryl chloride	75.2–85.3	
			[8]

	phosphorylation		
Bean starch	Acetylation		

Estimation of resistant starch levels in diverse food materials based on different processing methods and expressed in terms of percentage. Copyright obtained from Tamura et al. [39].

**Table 3 nutrients-11-01497-t003:** The percentage of starch hydrolysis, hydrolysis index (HI), and estimated glycemic index (EGI) for homogenized and non-homogenized heated rice samples with different degrees of heating during a virtual in vitro ingestion process.

		C∞ (%)	k × 10^−2^ (min^−1^)	HI	eGI
**Homogenized**	R	41.0 ± 3.7 c	4.51 ± 0.54 b	39.3 ± 2.5 d	61.3 ± 1.4 d
	MH	69.0 ± 4.6 ab	19.11 ± 1.26 a	78.0 ± 5.1 b	82.5 ± 2.8 b
	CC	78.1 ± 1.0 a	22.25 ± 6.94 a	88.5 ± 2.1 a	88.3 ± 1.1 a
**Non-Homogenized**	R	8.6 ± 1.4 d	3.07 ± 0.52 b	8.1 ± 1.0 e	43.8 ± 0.4 e
	MH	63.2 ± 9.1 b	3.03 ± 0.49 b	54.1 ± 4.8 c	69.4 ± 2.6 c
	CC	66.3 ± 2.2 b	3.28 ± 0.67 b	58.4 ± 3.1 c	71.8 ± 1.7 c

Different letters in the same column indicate significant differences (*p* < 0.05) (*n* = 3). R, raw; MH, moderately heated; CC, completely cooked; C∞, equilibrium starch hydrolysis percentage; k, kinetic constant. Copyright obtained from Tamura et al. [39].

**Table 4 nutrients-11-01497-t004:** Resistant starch level in cereal starches after processing.

Starch Source	Treatment	RS	Method of RS Analysis	Reference
Japonica brown rice	Pre-soaking in water at 30 or 40 °C to reach 25% humidity + heated	30.2–30.4%	[48]	[8]
	Pre-soaked in water at 20 or 45 °C to reach 30% humidity + heated	21.7–27.9%		
Waxy rice	Natural waxy rice starch	10.30%	[48]	[76]
	Gelatinized starch	3.00%		
	Ascetically gelatinized starch at 60 °C for 5 min	8.60%		
	Partially gelatinized starch at 70 °C for 5 min	7.70%		
Pastry wheat flour	Extracted at 20% humidity; 150/200/250 rpm; 40–120 °C; stored at 4 °C/0 days	0.48–0.52%	Megazyme^®^ assay	[77]
	Extruded at 20% humidity; 150/200/250 rpm; 40–120 °C; stored at 4 °C/7–14 days	1.21–1.35%		
	Extracted at 40% humidity; 150/200/250 rpm; 40–120 °C; stored at 4 °C/0 days	0.63–0.67%		
	Extracted at 40% humidity; 150/200/250 rpm; 40–120 °C; stored at 4 °C/7–14 days	1.52–1.86%		
	Extracted at 60% humidity; 150/200/250 rpm; 40–120 °C; stored at 4 °C/0 days	2.54–2.65%		
	Extracted at 60% humidity; 150/200/250 rpm; 40–120 °C; stored at 4 °C/7–14 days	3.55–4.25%		
Corn	Acid modified with 1.64 M HCl at 40 °C for 4 h + gelatinized + sterilized at 121 °C for15 min + freeze dried	5%	AOAC 991.43 [36]	[78]
	Acid reformed with 1.64 M Hydrochloric acid at 40 °C for 4 h + gelatinized + sterilized at 121°C for15 min + Stored at 95 °C for 48 h + freeze dried	12%		
Corn	Normal corn starch	19.70%	modified as per [79]	[76]
	Galvanized with high moisture for 24 h at 50 °C	18.30%		
	Moist–heat management (30% moisture + 24 h at ambient temperature + 120 °C for 24 h)	16.90%		
	Galvanized with high humidity for 24 h at 50 °C + Moist–heat treatment (30% humidity + 24 h ambient temperature + 120 °C for 24 h)	17.30%		
	Moist–heat treatment (30% moisture + 24 h ambient temperature + 120 °C for 24 h) + annealed with excess water for 24 h at 50 °C	19.70%		
High-amylose corn	Sterilized at 121 °C + Stored at 4 °C for 24 h (repeated twice)	30%	[36]	[36]
	Sterilized at 121 °C + Stored at 4 °C for 24 h (repeated twice) + hydrolyzed with 0.1 M organic acid	39%		

Resistant starch levels in different cereal-based starches after processing and expressed in terms of percentage. Copyright obtained from Tamura et al. [39].

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
