# Peer review of "Effect of Rice Processing towards Lower Rapidly Available Glucose (RAG) Favors Idli, a South Indian Fermented Food Suitable for Diabetic Patients"

_nutrients, 2019, doi:10.3390/nu11071497_

Reviewer 1 Report

Manuscript Review Comments

Title: “Effect of Rice Processing on Digestibility towards Diabetes Patients” (nutrients-508938)

Chelliah et al. present a review on the effect of different rice derived products of different origin in diabetic patients. The manuscript presents several problems including an urgent need for English grammar and style revision. Also, after checking for plagiarism, the article has shown a significant overlap of more than 41% with other related publications. Specifically, figures 3, 5 and 6 as well as table 4 are already published by Perera A et al. (2010) in Food Research International (Resistant starch: A review of analytical protocols for determining resistant starch and of factors affecting the resistant starch content of foods) and figure 4 from Maler TV et al. (2017) in mBio Journal (Impact of Dietary Resistant Starch on the Human Gut Microbiome, Metaproteome, and Metabolome). These are the most significant examples but this is also present throughout several paragraphs and sections of the text, as well as figure footnotes.

Author Response

Response to Reviewer 1 Comments

Ms. Ref. No.: 508938

Title: Effect of Rice Processing on Digestibility towards Diabetes Patients

Journal: Nutrients

Comments from the editor:

Please revise the manuscript according to the reviewers' comments and upload the revised file within 10 days.

-       The authors are grateful to the editor and reviewers for careful review of the manuscript and constructive comments. The manuscript was revised based on these comments. All changes in the revised manuscript are highlighted in red text font. The language of the research article is edited by a native English speaker and proof read. A point-by-point response to comments is included below.

Reviewer's comments:

Reviewer #1:

Manuscript Review Comments

Chelliah et al. present a review on the effect of different rice derived products of different origin in diabetic patients. The manuscript presents several problems including an urgent need for English grammar and style revision. Also, after checking for plagiarism, the article has shown a significant overlap of more than 41% with other related publications.

-       The overlap were corrected all over the manuscript, further in addition the he minor error and grammar have been corrected - The language of the research article is edited by a native English speaker and proof read.

-       Specifically, figures 3, 5 and 6 as well as table 4 are already published by Perera A et al. (2010) in Food Research International (Resistant starch: A review of analytical protocols for determining resistant starch and of factors affecting the resistant starch content of foods) and figure 4 from Maler TV et al. (2017) in mBio Journal (Impact of Dietary Resistant Starch on the Human Gut Microbiome, Metaproteome, and Metabolome). These are the most significant examples but this is also present throughout several paragraphs and sections of the text, as well as figure footnotes.

-       We are highly grateful to the reviewer for giving remarks on the article and precious time.

As per reviewer valuable suggestion, we had rearranged and elaborated the order of the titles and subheading with description inside the manuscript, based on the figures and tables, the copyright has been obtained from the following published manuscript.

-       In addition the duplication of the sentence has been edited throughout the manuscript.

References

1.        Impact of the degree of cooking on starch digestibility of rice – An in vitro study

2.        A review of analytical protocols for determining resistant starch and of factors affecting the resistant starch content of foods

Reviewer 2 Report

This review “Effect of Rice Processing on Digestibility towards  Diabetes Patients” described the Idli / Dokala one of mainly developed from rice is a fermented rice item due to its high edibility property and easily digestibility nature it increases post-prandial glucose levels and higher glycemic index. It is suggested that altering RAG to SAG have beneficial towards diabetes.

The review article is impressive; however, it can be improved by incorporating the following comments. I have the following concerns.

1. The title of this review is confusing ‘Effect of Rice Processing on Digestibility towards Diabetes Patients.’ The readers would assume this review will discuss the relationship of rice processing on digestibility towards diabetes subjects. Because it is known that rice has a higher glycemic index hence not suggestive to diabetes, as also shown in the text. Next, the readers would expect this review will discuss the rationales of intestinal digestion/absorption.  Hence I would suggest improving the title more specific to focus on the review. Say, for example, “Rice processing towards lower Rapidly Available  Glucose (RAG) favors South Indian fermented food Idli suitable for diabetic patients.”

2. The affiliation for the 5th is missing, or number is not placed, in the same way, *, ** for corresponding and co-corresponding messed up. It must be corrected.

3. the Idlis made with fermented rice and black lentils (urad dal), idlis are a great source of carbohydrates, and it is steamed; south Indian. Dhoklas are steaming a fermented batter of gram flour and yogurt. It is probiotic and has high nutritive value with the low glycemic index. The authors must correct throughout the manuscript starting from abstract Idli/Dokala to Idli. Since both are different and the preparation and the Dokala is not rice-based food, unlike idli.

4. It is good to introduce the review points not only diabetes aspect. Since review about rice processing with low RAG and focus on idli. Hence, it is must to incorporate those stuff in the introduction. It is not an introduction to only diabetes. This section must be rewritten to introduce the review aspects.

5. Table 1.  Multiple references need else source must be mentioned.

6.Figure 7.  Is that original experiment done for this review paper or else published data if so provide a reference for that?

7. Line 213-216. Figure 7 is coming before Figure 6. It is always good to provide sequence manner. If it needed one may quote the same figure later also.  Reframe the sentence to make in sequence manner or swap figure if the text goes in that way.

8. The conclusion is written poorly. It must be improved say, for example, summarize the review and conclude firmly.

9. Check for abbreviations which are not abbreviated throughout the manuscript and first instance it should be abbreviated even in the abstract.

10. As such, each paragraph looks fine but not logically connected to each portion. I suggest working on logical connect each portion of the review to make more cohesive of the content.

10. There are many other minor errors of syntax and grammar throughout the text, which need to be fixed.

Author Response

Response to Reviewer 2 Comments

Ms. Ref. No.: 508938

Title: Effect of Rice Processing on Digestibility towards Diabetes Patients

Journal: Nutrients

Comments from the editor:

-       If the reviewers have suggested that your manuscript should undergo extensive English editing, please address this during revision. We suggest that you have your manuscript checked by a Native English speaking colleague or use a professional English editing service.

-       The authors are grateful to the editor and reviewers for careful review of the manuscript and constructive comments. The manuscript was revised based on these comments. All changes in the revised manuscript are highlighted in red text font. The language of the research article is edited by a native English speaker and proof read. A point-by-point response to comments is included below.

Reviewer's comments:

Reviewer #2:

-        This review “Effect of Rice Processing on Digestibility towards Diabetes Patients” described the Idli / Dokala one of mainly developed from rice is a fermented rice item due to its high edibility property and easily digestibility nature it increases post-prandial glucose levels and higher glycemic index. It is suggested that altering RAG to SAG have beneficial towards diabetes.

-        The review article is impressive; however, it can be improved by incorporating the following comments. I have the following concerns.

-        We are thankful to the reviewer for providing valuable comments on the article and also for your precious time. The comments and suggestions raised have been addressed in the article. The response for individual comments is given below.

-        1. The title of this review is confusing ‘Effect of Rice Processing on Digestibility towards Diabetes Patients.’ The readers would assume this review will discuss the relationship of rice processing on digestibility towards diabetes subjects. Because it is known that rice has a higher glycemic index hence not suggestive to diabetes, as also shown in the text. Next, the readers would expect this review will discuss the rationales of intestinal digestion/absorption.  Hence I would suggest improving the title more specific to focus on the review. Say, for example, “Rice processing towards lower Rapidly Available Glucose (RAG) favors South Indian fermented food Idli suitable for diabetic patients.”

-         (Line 1-3) : As per the reviewer valuable comment, we had changed the title of the manuscript “Effect of Rice processing towards lower Rapidly Available Glucose (RAG) favors South Indian fermented food Idli suitable for diabetic patients” based on the focus of the manuscript.

-        2. The affiliation for the 5th is missing, or number is not placed, in the same way, *, ** for corresponding and co-corresponding messed up. It must be corrected.

-       (Line 13-17): As per the reviewer indicated, the affiliation and corresponding authorship has been corrected.

-        3. The Idlis made with fermented rice and black lentils (urad dal), idlis are a great source of carbohydrates, and it is steamed; south Indian. Dhoklas are steaming a fermented batter of gram flour and yogurt. It is probiotic and has high nutritive value with the low glycemic index. The authors must correct throughout the manuscript starting from abstract Idli/Dokala to Idli. Since both are different and the preparation and the Dokala is not rice-based food, unlike idli.

-         (Line 24, 26, 30) As per the reviewer valuable suggestion the throughout the manuscript the dokala was changed as rice based prepared dhokla variety and idli were discussed as food value towards the resistant starch

-        4. It is good to introduce the review points not only diabetes aspect. Since review about rice processing with low RAG and focus on idli. Hence, it is must to incorporate those stuff in the introduction. It is not an introduction to only diabetes. This section must be rewritten to introduce the review aspects.

-        As per the reviewer valuable comment, we had incorporated the review information on rice processing on idli product. (Line 50-71): Introduction were rearranged  based on rice processing with low RAG and focus on idli

-         The popular traditional rice based Indian food such as Idli, rice based Dokala mainly prepared with parboiled rice, which is rich in amylose content leads to soft and spongy textural characteristics [4]. Fermentation and cooking causes reduction in 40% of oligosaccharides and results in reduction in levelness of the product and based on the substitution of other grains such as pulses (kidney beans, black gram, broad bean, chick pea, soybean, lentil), hence depends on the composition the starch digestibility differs indirectly responsible to understand the context of type 2 diabetes [5]. Several studies were conducted to understand the blood glucose response based on the carbohydrate content of the food product, usually the blood glucose were quantified based on glycemic index (GI). Apart from glycemic index, the resistant starch (RS) content measured based on starch digestibility.  The fabrication of gradually digestible starch (GDS) and RS has been establish to alleviate the quantity of blood sugar and reveal the aids to human health [6]. The RS level of rice were exaggerated based on the heat treatment methods reported that high pressure heating (116°C, 10min) improved the RS level in raw rice from 6.29% into 14%. Nevertheless, the parboiling abridged the RS by 50% in lengthy slim rice. It appeared that heating at more than 100 °C possibly enhance RS content. The moist heat treatment (MHT) considered as hydrothermal method, which change the physiochemical properties of starch and is mostly steered at controlled humidity level (lower than 35%) and heating at (84–140 °C) within 15 min to 16 h [7]. Leads to gelatinization of the rice product based on lower moisture content. In addition, change’s, such as development of complexes of amylose-lipid (ALC) and lowered in enzymatic vulnerability, occur during MHT [8]. The quantity of RS in MHT treated starch leads to break down of granule minor structure. The assembly of gradually digestible starch (GDS) and RS has been initiated to regulate the quantity of blood sugar and parade towards welfares of health [6].

-         References

1.        Kannan, D., Chelliah, R., Vinolya Rajamanickam, E., Srinivasan Venkatraman, R., & Antony, U. (2015). Fermented batter characteristics in relation with the sensory properties of idli. Hrvatski časopis za prehrambenu tehnologiju, biotehnologiju i nutricionizam, 10(1-2), 37-43.

2.        Chelliah, R., Ramakrishnan, S. R., Premkumar, D., & Antony, U. (2017). Accelerated fermentation of Idli batter using Eleusine coracana and Pennisetum glaucum. Journal of food science and technology, 54(9), 2626-2637.

3.        Lehmann, U., & Robin, F. (2007). Slowly digestible starch–its structure and health implications: a review. Trends in Food Science & Technology, 18(7), 346-355.

4.        Arns, B., Bartz, J., Radunz, M., do Evangelho, J. A., Pinto, V. Z., da Rosa Zavareze, E., & Dias, A. R. G. (2015). Impact of heat-moisture treatment on rice starch, applied directly in grain paddy rice or in isolated starch. LWT-Food Science and Technology, 60(2), 708-713.

5.        Chung, H. J., Liu, Q., & Hoover, R. (2009). Impact of annealing and heat-moisture treatment on rapidly digestible, slowly digestible and resistant starch levels in native and gelatinized corn, pea and lentil starches. Carbohydrate Polymers, 75(3), 436-447.

6.        Hu, F. B., Van Dam, R. M., & Liu, S. (2001). Diet and risk of type II diabetes: the role of types of fat and carbohydrate. Diabetologia, 44(7), 805-817.

7.        Ley, S. H., Hamdy, O., Mohan, V., & Hu, F. B. (2014). Prevention and management of type 2 diabetes: dietary components and nutritional strategies. The Lancet, 383(9933), 1999-2007.

-        5. Table 1.  Multiple references need else source must be mentioned.

-        (Line 170-171): As per the reviewer valuable comment, quantification of total starch, total dietary fiber, resistant starch level in diverse raw food material and expressed in terms of percentage, Copyright were obtained from Tamura et al [77]

-        (Line 259-260): Table 2. Preparation of resistant starch in starch rich products by different treatments.

-       (Line 287-292): Table 3.The percentage of starch hydrolysis, hydrolysis index (HI) and valued glycemic index (EGI) for homogenized and non-homogenizedheated rice samples with different degrees of heating during virtual in vitro ingestion process. Copyright were obtained from Tamura et al [77]

-       (Line 335- 337): Table 4. Resistant starch level in cereal starches as prejudiced by processing. Copyright were obtained from Tamura et al [77]

-        Reference

-         Tamura, M., Singh, J., Kaur, L., & Ogawa, Y. (2016). Impact of the degree of cooking on starch digestibility of rice–An in vitro study. Food chemistry, 191, 98-104.

-        6. Figure 7.  Is that original experiment done for this review paper or else published data if so provide a reference for that?

-        (Line 244-245): Figure 7. The transmission electron microscopic image indicates the effect of heat treatment in the rice prepared Idli product with the control Idli (No-Heat Treated).  (Un-published data) Research work is ongoing currently in the laboratory (Heat treatment and physiochemical variations were documented).

 -        7. Line 213-216. Figure 7 is coming before Figure 6. It is always good to provide sequence manner. If it needed one may quote the same figure later also.  Reframe the sentence to make in sequence manner or swap figure if the text goes in that way.

-        As per reviewer’s valuable recommendation  - Figure 6 Indicated in the text  – (Line 197) subsequently Figure 6 represented in (Line 200-205), further Figure 7 indicated in the text – (Line 239) subsequently Figure 7 represented in (Line 242-245)

-        8. The conclusion is written poorly. It must be improved say, for example, summarize the review and conclude firmly.

-        As per reviewer’s valuable recommendation , (Line- 33-338) the conclusion were reprised and summarized as follows

-        Conclusion

The review evidently reveals that a slight amendment of rice treating can modify the physiochemical properties of rice. This rice plugs the bulk that is, to satiety but most being not absorbed doesn’t leads to increase in blood glucose level. Additional effect of fiber and further effects as discussed, over-all and (Low-density lipoprotein) LDL dietary fat was abridged. The process is economically affordable and not expensive. The only disadvantage is minor different in taste and raised fuel rate on heating rice. Based on the review it was concluded that the thermal treatment of rice is a better method for increasing resistant starch content which in turn may be useful for diabetes patients. Hence heat treatment of rice is preferred for the preparation of low GI Idli and other rice based food products. 

-        9. Check for abbreviations which are not abbreviated throughout the manuscript and first instance it should be abbreviated even in the abstract.

-        As per the reviewer suggestion these sentences, it be modified and rephrased.

-        (Line 28, 131, 136, 134, 219)- RAG and SAG values, SEM analysis

-        (Line 58, 59,64,69, 149, 175,176) – Resistance starch – RS

-        (Line 64, 68, 69, 252,254) –  Moist heat treatment - MHT

-        (Line 29,339) - glycemic index (GI)

-        (Line 59,70,258, 259)- gradually digestible starch (GDS)

-        (Line 113, 116, 270, 272)- American Diabetes Association (ADA)

-        10. As such, each paragraph looks fine but not logically connected to each portion. I suggest working on logical connect each portion of the review to make more cohesive of the content.

-        As per reviewer’s recommendation we had rearranged the review work

-        1. Introduction  (Line -35-70)

o    Figure 1. Development of cereal based products for diabetes patients.

-        2. Dietary Factors and type 2 diabetes (Line -71-77)

-        3. Contemporary perceptions towards deterrent level of Starch/carbohydrate/Fiber/Fat in Diet (Line -78-83)

-        4. Rice and glycemic index mechanism (Line -84-95)

o    Figure 2. Digestive enzymes secreted by different organs contributing in the digestion of foods in humans. The digestive system, an extension of the environment into the body, also stashes mucus, acid, bicarbonate and salts to enable hydrolysis of foods and absorption of nutrients essential for life.

-        5. Glycemic Index and Glycemic Load (Line -104-129)

-        6. Role of Rag and Sag on diabetic patients (Line -130-141)

o    Figure 3. Goñi et al. [75], Perera [78] procedure on the determination of resistant starch (RS) in nourishments. Copyright were obtained from Perera et al

-        7. Resistant Starch on gut microbiota modulation (Line -147-188)

o   Figure 4. Summary of distinguished enzymes, subsequent pathways, selective species, and specific metabolites that were expressively obstructed through resistant starch diet.      Blue arrows/frames, increased in HRS; orange frames, decreased in HRS; black arrows, not detected or not increased in HRS over baseline; green arrows/frames, increased in LRS. 1, starch and metabolism of surcose; 2, glucose to pyruvate leads to glycolysis pathway; 3, 3-oxoacyl-(acyl carrier protein) synthase; 4, acetyltransferase based acetyl-CoA; 5, 3-hydroxyacyl-CoA dehydrogenase; 6, enoyl-CoA hydratase; 7, carrier protein enoyl-(acyl) reductase (NADH); 8, acetate transferase -CoA; 9, butyrate kinase; 10, synthase- citrate; 11, hydratase aconitate; 12 and 13, dehydrogenase isocitrate; 14, Ferredoxin oxidoreductase-2-ketoglutarate; 15, CoA-synthetase-succinyl; 16, fumarate reductase/succinate dehydrogenase; 17, fumarate hydratase; 18, dehydrogenase malate; 19, humanoid enzymes.

o    Table 1. Estimated total dietary fibre, total starch and resistant starch of some nourishment sources (g/100 g as eaten).

o    Figure. 5. Major steps according to Englyst et al. [40] protocol towards grit of enzyme resistant starch (RS3). The manifestation of an enzyme resistant fraction of starch in foods during the measurement of non-starch polysaccharides in foods. Copyright were obtained from Perera et al [78]

-        8. RS, Insulin, Glucose metabolism, GL and GI (Line -188-199)

o    Figure. 6. Instantaneous procedure for the determination of total starch (TS) as per Englyst et al. [62]. The freely available glucose (FG) and over-all glucose (TG) matters were restrained to quantify the over-all starch content of the sample. Copyright were obtained from Perera et a.

-        9. Fermented Foods – Idli (Line -205-214)

-        10. Limitation of Glycemic Index fall over 77 ± 2 considered as high and the Glycemic Load over 40.04 were considered as hiked. (Serving size: 250 grams) (Line -215-220)

-        11. Processing Method (Line -222-228)

-        12. Physical Treatment (Line -229-241)

o   Figure 7. The transmission electron microscopic image indicates the effect of heat treatment in the rice prepared Idli product with the control Idli (No-Heat Treated).  (Un-published data)

-        13. Microstructure of rice grains (Line -245-258)

o    Table 2. Preparation of resistant starch in starch rich products by different treatments.

-        14. Glycemic Index (Line -262-289)

o    Table 3. The percentage of starch hydrolysis, hydrolysis index (HI) and valued glycemic index (EGI) for homogenized and non-homogenized heated rice samples with different degrees of heating during virtual in vitro ingestion process.

-        15. Rapidly Available Glucose (RAG) and Slowly Available Glucose (SAG) (Line -290-308)

-        16. Processing Method and Resistant Starch (Line -309-329)

o    Table 4. Resistant starch level in cereal starches as prejudiced by processing.

-        17. Conclusion (Line -330-339)

-        10. There are many other minor errors of syntax and grammar throughout the text, which need to be fixed - English editing

-        Based on the reviewer valuable comments. The minor error and grammar have been corrected - The language of the research article is edited by a native English speaker and proof read.

Round  2

Reviewer 2 Report

I appreciate the authors for the extensive modification of the review manuscript as per my suggestions and comments. As of present format, in my opinion, it was satisfactory.